# Multi-View Graph Neural Networks with Language Models for Multi-Source Recommender Systems

## Abstract

Graph Neural Networks (GNNs) have become increasingly popular in recommender systems due to their ability to model complex user-item relationships. However, current GNN-based approaches face several challenges: They primarily rely on sparse user-item interaction data, which can lead to overfitting and limit generalization performance. Moreover, they often overlook additional valuable information sources, such as social trust and user reviews, which can provide deeper insights into user preferences and enhance recommendation accuracy. To address these limitations, we propose a multi-view GNN framework that integrates diverse information sources using contrastive learning and language models. Our method employs a lightweight Graph Convolutional Network (LightGCN) on user-item interactions to generate initial user and item representations. We use an attention mechanism for the user view to integrate social trust information with user-generated textual reviews, which are transformed into high-dimensional vectors using a pre-trained language model. Similarly, we aggregate all reviews associated with each item and use language models to generate item representations for the item view. We then construct an item graph by applying a meta-path to the user-item interactions. GCNs are applied to both the social trust network and the item graph, generating enriched embeddings for users and items. To align and unify these heterogeneous data sources, we employ a contrastive learning mechanism that ensures consistent and complementary representations across different views. Experimental results on multiple real-world datasets such as Epinions, Yelp, and Ciao demonstrate significant performance improvements over state-of-the-art methods.

## 1 Introduction

Recommender systems are powerful techniques widely adopted in various domains, including e-commerce platformsWang et al. (2020), online advertisingGharibshah and Zhu (2021), and video streaming services Liu et al. (2019). The main task of recommender systems is to predict whether a user will interact with a specific item. Collaborative filtering (CF) methods are among the most successful approaches in recommender systems, effectively predicting how likely a user is to interact with specific items Wang et al. (2019b). However, collaborative filtering (CF) methods face challenges like data sparsity and the cold start problem. Data sparsity occurs when there are limited interactions between users and items, making it difficult to generate accurate recommendations. GNNs have significantly succeeded in recommender systems due to effectively capturing complex user-item interactionsChen et al.He et al. (2020). Unlike traditional methods, GNNs can model direct and indirect relationships by leveraging the graph structure of user-item interactions. This allows them to learn rich, high-quality representations that account for collaborative patterns across the entire network. GNNs have successfully addressed challenges such as data sparsity and the cold start problem, leading to more accurate and personalized recommendations in domains like e-commerce, social media, and content streaming.

Neural Graph Collaborative Filtering (NGCF)Wang et al. (2019b) and LightGCNHe et al. (2020) are examples of successful GNN-based recommender models that have made significant contributions

to improving recommendation accuracy. These models use graph convolution to capture local information and aggregate data from neighboring nodes, incorporating collaborative signals into user and item representations. GNNs in recommender systems rely heavily on user-item interactions, which are often sparse, leading to several challenges, such as limiting the model's ability to learn meaningful patterns, propagating noisy information, which reduces the accuracy, over-smoothing, and scalability issues. To address these issues, contrastive learning has recently been employed to provide more robust and high-quality representations for users and items Yu et al. (2023). By leveraging both positive interactions (e.g., items a user interacts with) and negative samples (e.g., items they don't interact with), contrastive learning helps the model differentiate between similar and dissimilar user-item interactions, resulting in more discriminative embeddings and improving the overall recommendation accuracy, even in sparse data environments. However, additional sources of information, such as user social interactions and user reviews, are available in the form of unstructured text and can be utilized to improve the accuracy of GNN-based recommender systems. *The question is how to effectively integrate multi-source information, like user social data and reviews, into GNN-based models. Specifically, since user reviews are unstructured, how can we leverage them to extract meaningful semantic information and combine this data with GNNs to enhance recommendation performance?*

**Present work.** The present work proposes a multi-view graph representation learning strategy designed explicitly for recommender systems, addressing the limitations of traditional methods that often rely on sparse user-item matrices. By integrating multiple sources of information, such as user trust relationships and user reviews, this approach creates a more robust and informative framework. Incorporating user trust networks captures implicit connections between users, significantly enhancing recommendation accuracy by factoring in social influences on user preferences. Additionally, the use of a pre-trained language model like BERT allows the system to transform user-generated reviews into meaningful embeddings, leveraging the nuanced sentiments expressed in these texts to enrich user and item representations. A contrastive learning mechanism aligns these diverse representations, ensuring consistency and complementarity among the various data sources, which fosters a cohesive understanding of user preferences and item characteristics. By embracing this multi-view perspective, the method effectively captures the complexities of user behavior and item attributes, offering a more comprehensive solution compared to existing graph neural network-based recommenders that typically focus on a singular view. The key properties of the proposed method are listed as follows:

• The proposed method incorporates trust relationships and user reviews alongside traditional user-item interactions, enriching the data landscape for representation learning.

• By utilizing pre-trained language models like BERT for review embeddings, the method uses advanced natural language processing techniques to gain deeper insights into user sentiments and preferences.

• The contrastive learning approach used in the method helps integration of data taken from various sources.

• The integration of user trust dynamics enables the model to adapt to the social context of interactions, providing a more personalized recommendation experience.

## 2 RELATED WORKS

**GNN-based recommender systems:** GNNs have emerged as a powerful approach in graph-based recommender systems by effectively capturing complex relationships within user-item networksTang et al. (2008). These networks leverage the inherent graph structure of interactions to learn meaningful patterns. Typically, GNNs operate using the message-passing framework, where information is propagated and aggregated across multiple layers. This process allows the model to incorporate local and global information from neighboring nodes, enabling it to learn more expressive and rich representations for users and items. GNNs model user-item interactions as a graph to generate user and item embeddings by leveraging cross-layer information propagation. This process allows the model to capture both local and global patterns in the graph, as information from neighboring nodes is propagated through multiple layers. By aggregating this information, GNNs can produce more robust and informative embeddings for users and items, leading to improved recommendation accuracy

(Chen et al. Wang et al. (2019b)). Some studies, such as GraphRec Fan et al. (2019) and KCGN Huang et al. (2021), have utilized user social interactions to enhance recommendation performance by incorporating the influence of social connections into GNN-based models. Other models, such as KGAT Wang et al. (2019a), extend GNNs to operate on knowledge graphs, which provide rich, structured semantic information about entities and their relationships.

**Contrastive Self-Supervised Learning on Graphs:** Contrastive learning is a widely used self-supervised technique that leverages both positive and negative samples to learn discriminative features from data. By contrasting similar (positive) pairs with dissimilar (negative) ones, this approach helps models differentiate between meaningful patterns, leading to more robust and effective representations. In the context of recommender systems, a positive pair typically consists of a user and an item with which they have interacted (e.g., a purchase, rating, or click), while a negative pair consists of a user and an item they have not interacted with. Recent studies show contrastive learning has gained attention for its ability to address challenges such as data sparsity and noisy interactions. Some methods, such as HGCL Chen et al. (2023), propose a contrastive multi-view model that utilizes meta-path to capture multi-views of heterogeneous graphs. In Zou and Wang (2023), a contrastive learning model was proposed to learn representations by contrasting a heterogeneous graph with a meta-path-based homogeneous graph extracted from the heterogeneous graph. The approach leverages the structural differences between these two types of graphs. By applying contrastive loss, the model aims to capture meaningful distinctions between the homogeneous and heterogeneous graphs, thereby learning richer and more informative representations that reflect local and global data patterns. SimGRACE Xia et al. (2022) is a self-supervised learning framework that introduces contrastive learning by generating contrastive views through perturbations applied directly to the GNN encoder. Unlike traditional methods that create contrastive views by augmenting the input data (such as node features or graph structures), SimGRACE perturbs the GNN model parameters themselves to create different perspectives of the same graph. Zou and Wang (2023) employs contrastive learning to integrate the embeddings generated from user-item interactions with those obtained from homogeneous graphs constructed using meta-paths.

## 3 PRELIMINARIES

**Graph Convolutional Networks (GCNs)** Kipf and Welling (2016) are designed to operate directly on graph-structured data, allowing for effective representation learning from the connections between nodes. In the context of a graph, the input consists of an adjacency matrix A that encodes the relationships between nodes, and a feature matrix X that contains the features of each node. The fundamental operation in a GCN is performed through a series of graph convolutional layers. The update rule for node embeddings at layer $l$ is defined as follows:

$$Z^{(l+1)} = \sigma\left(\tilde{A} Z^{(l)} W^{(l)}\right). \tag{1}$$

where $Z^{(l)}$ represents the node embeddings at layer $l$ (for layer 0, $Z^{(0)}$ is the input feature matrix), $W^{(l)}$ is the trainable weight matrix at layer $l$, $\sigma$ is a non-linear activation function (e.g., ReLU), and $\tilde{A}$ is the normalized adjacency matrix defined as $\tilde{A} = \hat{D}^{-1/2}\hat{A}\hat{D}^{-1/2}$. Here $\hat{D} = A + I$ is the adjacency matrix with self-loops (adding the identity matrix I to include each node's own feature in the aggregation). $\hat{D}$ is the degree matrix of $\hat{A}$, where $\hat{D}_{ii} = \sum_j \hat{A}_{ij}$ and $\hat{D}^{-1/2}$ is the normalized degree matrix (square root inverse of the degree matrix). This normalization ensures that the node features are scaled properly, preventing exploding or vanishing gradients during training.

**LightGCN (Lightweight Graph Convolutional Network)** He et al. (2020) a graph-based approach designed for recommender systems, focusing on efficient representation learning from user-item interactions. The core idea in LightGCN is to leverage the graph structure of user-item interactions while simplifying traditional GCNs. The user-item interaction graph can be represented as an adjacency matrix A, where each entry $D_{ii}$ indicates the interaction between user $u_i$ and item $i_j$. The update rules for user embedding at layer $l$ is obtained as follows $Z_U^{(l)} = A_U Z_U^{(l-1)} + A_I Z_I^{(l-1)}$, where $A_U$ represents the user-user adjacency matrix, capturing implicit relationships between users, derived from the user-item interaction matrix A as $A_U = A \cdot A^T$. Similarly, the update rules for item embedding at layer $l$ is obtained as $Z_I^{(l)} = A_l Z_I^{(l-1)} + A_Z^{(l-1)}$, where $A_I$ is the item-item adjacency matrix derived from the user-item interaction as $A_I = A \cdot A^T$. After $L$ layers of graph

convolution, the final user and item embeddings are obtained by aggregating the outputs from all layers $Z_U = \sum_{l=1}^{L} Z_U^{(l)}$ and $Z_I = \sum_{l=1}^{L} Z_I^{(l)}$. The final recommendation score for a user-item pair is computed as the dot product of their corresponding embeddings $\hat{y} = Z_U \cdot Z_I$.

**BERT (Bidirectional Encoder Representations from Transformers)** Lee and Toutanova (2018) is a transformer-based model designed to generate contextual embeddings for text. It processes input sequences through several stages, starting with pre-processing, where the text is tokenized into sub-word units and encoded as numeric representations. The input text is tokenised, where each token is transformed into token IDs represented by the matrix $X = [x_1, x_2, \ldots, x_n]$, where each $x_i$ is a token ID. These IDs are mapped to dense vector representations using three types of embeddings: token embeddings, position embeddings (indicating the token's position), and segment embeddings (distinguishing between different sentences). The total input embedding for each token is calculated as $H^{(0)} = E_{\text{token}}(x_i) + E_{\text{position}}(i) + E_{\text{segment}}(s)$ where $s$ indicates the segment. BERT consists of multiple transformer layers, each employing self-attention and feedforward networks to refine the embeddings. The self-attention mechanism uses learned matrices for queries $Q$, keys $K$, and values $V$, computed from the input embeddings $H^{(l)}$ at layer $l$ as $Q = H^{(l)} W_Q$, $K = H^{(l)} W_K$, and $V = H^{(l)} W_v$. The attention scores are calculated as $Attention(Q, K, V) = softmax(QK^T/d)V$. Multi-head attention applies this process multiple times in parallel with different weight matrices and concatenates the results, followed by a linear transformation as $H'^{(l)} = MultiHead(H^{(l)})$. After self-attention, the output is passed through a position-wise feedforward network as:

$$H^{(l+1)} = ReLU(H'^{(l)} W_1 + b_1)W_2 + b_2 \tag{2}$$

where $W_1$, $W_2$ are weight matrices and $b_1$, $b_2$ are biases. After processing through all layers, the final output matrix $H^L$ contains contextual embeddings for each token, with $H^L = [h_1^{(L)}, h_2^{(L)}, \ldots, h_n^{(L)}]$, where each $h_i^{(L)}$ is a d-dimensional vector representing the contextual encoding of token $i$. Ultimately, BERT outputs the contextual embeddings as $H^{(L)} = BERT(X)$, providing rich representations for the input text.

# 4 METHODOLOGY

The proposed method is a multi-view graph GNN-based framework for recommender systems that integrates multiple sources of information such as user-item interactions, user trust networks, item profiles, and user reviews into a unified structure. This enables the system to capture different perspectives and enhance its predictive power. The framework consists of three distinct views named user-item view, user view, and item view. The overall architecture of the proposed framework is show in Figure 1.

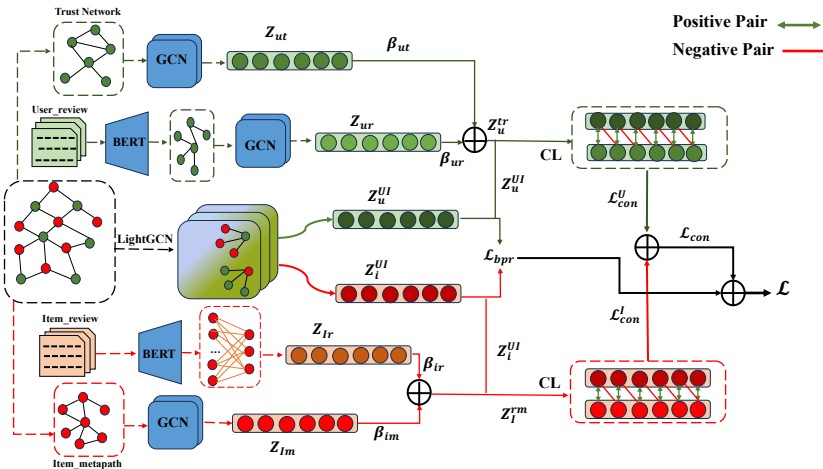

Figure 1: Overall structure of MvL-GNN.

**User-item view:** In this view, user-item interaction data is leveraged to capture the relationships between users and items. We follow the similar strategy proposed in He et al. (2020) to learn user embeddings $Z_U^{UI}$ and item embeddings $Z_I^{UI}$ from the user-item interaction graph. Let $A^{UI}$ is adjacency matrix of the user-item interaction graph where $A_{ij}^{UI} = 1$ indicates the interaction between user $u_i$ and item $i_j$. The update rules for user embedding at layer $l$ obtained as follows $H_U^{(l)} = A_U^{UI} H_U^{(l-1)} + A^{UI} H_I^{(l-1)}$, where $A_U^{UI}$ the user-user adjacency matrix, computed $A_U^{UI} = A_U^{UI} \cdot A_U^{UI^T}$. Similarly, the update rules for item embedding at layer $l$ is obtained as $H_I^{(l)} = A_I^{UI} H_I^{(l-1)} + A^{UI} H_I^{(l-1)}$, where $A_I^{UI}$ is the item-item adjacency matrix computed as $A_U^{UI} = A^{UI^T} \cdot A^{UI}$. After $L$ layers of graph convolution, the final user and item embeddings are obtained by aggregating the outputs from all layers $Z_U^{UI} = \sum_{l=1}^{L} H_U^{(l)}$ and $Z_I^{UI} = \sum_{l=1}^{L} H_I^{(l)}$. These embeddings encode latent features from historical interactions, allowing the system to understand user preferences and item characteristics.

**User view:** This view aims to combine multiple sources of information to generate a comprehensive user representation by leveraging both the user's trust network and their textual reviews. First, we apply the GCN to the user trust graph, where users are connected based on trust relationships. Let $G^{Tr} = (U, E)$ represents the trust graph, with U as the set of users and E as the trust relationships. The adjacency matrix $A_U^{Tr}$ of this graph and the initial feature matrix $X_u$ for users are used in a GCN, where each layer updates the user embeddings. The GCN's propagation rule for layer $l$ is given by:

$$H_u^{(l+1)} = \sigma \left( \hat{D}_U^{Tr-\frac{1}{2}} \hat{A}_U^{Tr} \hat{D}_U^{Tr-\frac{1}{2}} H_v^{(l)} W^{(l)} \right) \tag{3}$$

where $\hat{A}_U^{Tr} = A_U^{Tr} + I$, $\hat{D}_U$ is the degree matrix on $\hat{A}_U^{Tr}$, $W^{(l)}$ is a learnable weight matrix, and $\sigma(\cdot)$ is an activation function, such as ReLU. After applying $L$ layers, the final user trust embedding is $Z_U^{Tr} = H_U^{(L)}$. In addition to trust data, we utilize user reviews, which are textual data. These reviews are encoded using a pre-trained BERT model, transforming the reviews for user u into an embedding denoted as $E_u^R = BERT(Reviews_u)$. Since BERT embeddings are typically high-dimensional, we apply a linear transformation to reduce their size to match the dimensionality of $Z_U^T$, using the equation:

$$Z_U^R = W_R E_U^R + b_R \tag{4}$$

where $W_R$ is the weight matrix and $b_R$ is the bias term. To combine the user trust embedding $Z_U^T$ and the review embedding $Z_U^R$, we apply a node attention mechanism that assigns different weights to each component. Attention coefficients $\beta^{Tr}$ and $\beta^R$ are calculated for $Z_U^T$ and $Z_U^R$ as follows: $\beta^{Tr} = tanh(W^T \cdot [Z_u \| Z_r^T])$ and $\beta^R = tanh(W^R \cdot [Z_u \| Z_r^T])$, where $\|$ denotes concatenation and $W^T$ and $W^R$ are learnable weight matrices. The final user representation is a weighted combination of the two embeddings:

$$Z_U^{TrR} = \beta^{Tr} Z_U^{Tr} + \beta^R Z_U^R \tag{5}$$

This mechanism allows the model to balance the information from both the trust network and the review text based on their relative importance.

**Item view:** This view aims to generate item's representation by combining two key sources of information: a graph of item interactions and item reviews. We first construct a homogeneous item graph using an *m:I-U-I* metapath, where items are connected through shared users in the user-item interaction matrix. Let $A_I$ represent the adjacency matrix for the item graph, where the connections between items are based on interactions with the same users. It can be constructed from the original user-item interaction matrix as $A_I^{Me} = A_{UI}^T A_{UI}$. We then apply the GCN to learn item embeddings $Z_I^{Me}$. Let $X_I$ be the initial feature matrix of the items (this can be a one-hot encoded vector or any other feature representation). Similar to the user view, we propagate item features through the GCN using the normalized adjacency matrix $\hat{A}_I^{Me}$. The update rule for the GCN layers is:

$$H_I^{(l+1)} = \sigma \left( \tilde{A}_I^{Me} H_I^{(l)} W_I^{(l)} \right) \tag{6}$$

where $H_I^{(l)}$ is the item representation at layer $l$ with $H_I^{(0)} = X_I$, and $w_I^{(l)}$ is the learnable weight matrix for layer $l$. After applying $L$ layers, we obtain the final item embedding from the interaction

graph $Z_I^{Me} = H_I^{(L)}$. Next, similar to the user view, we consider the textual reviews associated with each item. For each item, we collect all user reviews and encode them using a pre-trained BERT model. Let the reviews for item $i$ be denoted as $Reviews_i$. We apply BERT to generate a textual embedding for these reviews $E_i^R = BERT(Reviews_i)$. The output of BERT is typically a high-dimensional embedding, so we reduce its dimensionality using a linear transformation to match the size of $Z_I^{Me}$. The reduced embedding from the reviews is given by $Z_I^R = W_R E_I^R + b_R$. The final embedding $Z_I^{MeR}$ is obtained by combing $Z_I^R$ using the same attention mechanism used in user-view.

**Training the model:** In this framework, we employ two types of loss functions: a supervised loss function and a self-supervised contrastive loss function, which are combined to train the model effectively. Supervised loss is based on Bayesian Personalized Ranking (BPR), which is commonly used in recommender systems. The goal of this loss is to model the relative preferences between items for each user. For a user $u_i$ and an item $i_j$, the model predicts the recommendation score using the dot product of their embeddings from the user-item interaction view:

$$\hat{y}_{ij} = Z_U^{UI}(i) \cdot Z_I^{UI}(j) \tag{7}$$

where $Z_U^{UI}(i)$ and $Z_I^{UI}(j)$ show the embeddings of user $u_i$ and item $i_j$ respectively. The actual recommendation label is denoted by $y_{ij}$. The BPR loss function aims to maximize the margin between observed interactions (i.e., recommended items) and unobserved interactions (i.e., non-recommended items). Mathematically, the BPR loss is formulated as:

$$\mathcal{L}_{bpr} = - \sum_{(i,j,k) \in D} \log \sigma \left( \hat{y}_{ij} - y_{ik} \right) \tag{8}$$

where $\sigma$ is the sigmoid function, D is the set of user $u_i$, positive item $i_j$, and negative item $i_k$ triplets. In addition to the supervised loss, we use a contrastive loss to align the two different embeddings for users and items. Each user and item has embeddings from two different views: the user-view/item-view and the user-item interaction view. The goal of the contrastive loss is to ensure that these different embeddings for the same user or item are consistent. For the user, we have two embeddings: $Z_U^{TrR}$ (the user embedding from the user-view) and $Z_U^{UI}$ (the user embedding from the user-item interaction view). The goal is to maximize the agreement between $Z_U^{TrR}(u)$ and $Z_U^{UI}(u)$ for the same user $u$, while minimizing the similarity between embeddings of different users. The contrastive loss for users is given as:

$$\mathcal{L}_{con}^U = - \sum_u \log \frac{\exp(\text{sim}(Z_U^{TrR}(u), Z_U^{UI}(u)/\tau)}{\sum_{u' \in U} \exp(\text{sim}(Z_U^{TrR}(u'), Z_U^{UI}(u^i))/\tau)} \tag{9}$$

where $\tau$ is the temperature parameter that controls the sharpness of the *softmax* distribution and $sim(.,.)$ is the similarity (often cosine similarity) between the user embeddings. Similarly, for items, we align the item embeddings from the item-view ($Z_I^{MeR}$) and the user-item interaction view ($Z_I^{UI}$) using the InfoNCE loss denoted as $\mathcal{L}_{con}^I$. The loss encourages item embeddings from the two views to be similar for the same item while distinguishing them from embeddings of different items. The total contrastive loss combines both user and item contrastive losses as $\mathcal{L}_{con} = \mathcal{L}_{con}^U + \mathcal{L}_{con}^I$. The total loss function is the sum of the supervised BPR loss and the contrastive loss:

$$\mathcal{L}_{con} = \mathcal{L}_{bpr} + \lambda \mathcal{L}_{con} \tag{10}$$

Here, $\lambda$ controls the relative weight of the contrastive loss, allowing the model to balance learning good representations from the supervised BPR loss and aligning the embeddings from different views using the InfoNCE loss. We use this loss functions to train the learnable parameters of the framework.

## 5 EXPERIMENTS

### 5.1 DATASETS AND PERFORMANCE MEASURES

To assess the performance of the MvL-GNN method, we conducted a series of experiments on three real-world datasets: Yelp, Ciao, and Epinions. The details of these datasets are shown in Table 1. We investigate the effectiveness of our model by conducting different experimental scenarios designed to

address the following key questions:

*Q1: In what ways does our model outperform existing state-of-the-art methods in terms of recommendation accuracy and robustness?*

*Q2: How does the performance of our model vary across different evaluation scenarios, such as varying data sparsity and cold start conditions?*

*Q3: What is the impact of hyperparameter tuning on the effectiveness and efficiency of the proposed method?*

Table 1: Real-world Dataset

| Datasets | Users# | Items# | Interactions# |
|----------|--------|--------|---------------|
| Ciao | 6776 | 101415 | 265308 |
| Epinions | 11734 | 23396 | 26517 |
| Yelp | 161305 | 114852 | 957923 |

## 5.2 COMPARISION WITH STATE-OF-THE-ART METHODS

In response to Q1, we performed a set of experiments to compare the proposed method (i.e. MvL-GNN) with a set of state-of-the-art methods including HAN Wang et al. (2019c), HeCoWang et al. (2021), HGT Hu et al. (2020), MHCN Yu et al. (2021), SMIN Long et al. (2021), HGCL Chen et al. (2023). Table 2 reports the comparison results for Ciao, Epinions and Yelp datasets in terms of HR(Hit Ratio) and NDCG (Normalized Discounted Cumulative Gain) metrics. The results indicate that MvL-GNN consistently outperforms all other models across all datasets and metrics, highlighting its potential to enhance recommender system performance significantly.

Table 2: Performance of different methods on three datasets

| Datas | Metrics | HAN | HeCo | HGT | MHCN | SMIN | HGCL | MvL-GNN |
|-------|---------|------|------|------|------|------|------|---------|
| Ciao | HR@10 | 0.6772 | 0.6867 | 0.6939 | 0.7053 | 0.7008 | 0.7310 | 0.7400 |
| | ND@10 | 0.4708 | 0.4469 | 0.4867 | 0.4869 | 0.4928 | 0.5199 | 0.5301 |
| Epinions | HR@10 | 0.7630 | 0.7998 | 0.8150 | 0.8201 | 0.8045 | 0.8323 | 0.8474 |
| | ND@10 | 0.5810 | 0.6026 | 0.6145 | 0.6158 | 0.6234 | 0.6376 | 0.6595 |
| Yelp | HR@10 | 0.7731 | 0.8359 | 0.8364 | 0.8344 | 0.8401 | 0.8743 | 0.8862 |
| | ND@10 | 0.5601 | 0.5838 | 0.5880 | 0.5800 | 0.6012 | 0.6295 | 0.6402 |

## 5.3 EVALUATION UNDER VARIOUS SCENARIOS

To address Q2, several experiments were conducted under various scenarios to verify that incorporating multiple views of users and items is essential for learning high-quality representations. Here, we primarily focus on the impact of source information in each user-view and item-view embedding on recommendation performance. Our framework considers multiple perspectives of user and item interactions (user-view, item-view, and the user-item interaction view). The user-view embedding combines the user's trust network with their interactions based on textual reviews. In contrast, the item-view embedding integrates a meta-path-based graph of item interactions and reviews. In this section, we aim to analyze the impact of each information source and its influence on extracting high-quality embeddings for conducting contrastive learning with the user-item view embedding in the proposed method. We evaluated the impact of user and item information and defined five evaluation scenarios to demonstrate the effectiveness of each aspect of our approach. Details of these scenarios are explained in Figure 2. The performance of our proposed method, along with the compared scenarios, is presented in Figure 2 for both Hit Ratio and NDCG metrics. From these results, we can conclude that the user's trust network plays a crucial role in the user-view embedding. Additionally, for the item-view embedding, the item's reviews significantly influence the item's representation. Finally, in the fifth scenario (MvL-GNN), we observed that the performance of our proposed method improves by incorporating both sources of information—user trust and item reviews—into the user-view and item-view embeddings.

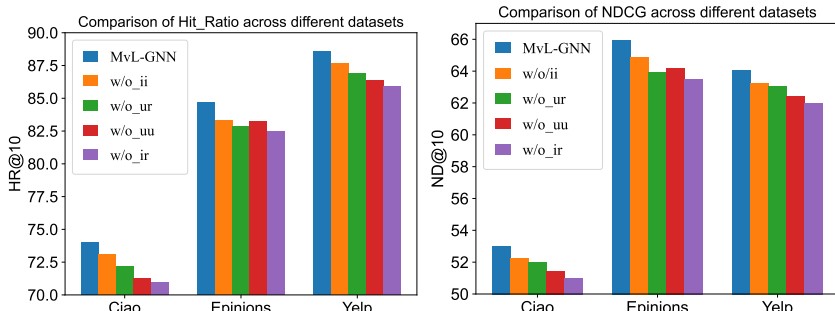

Figure 2: Evaluation of MvL-GNN across different scenarios. **w/o_ir:**We trained the model without using the item representations extracted by the transformer model, meaning that the item review information was removed from the item view.**w/o_ii:** In this scenario, we removed the meta-path-based graph and used only the item's reviews to extract the item-view embedding.**w/o_ur:** In this scenario, we excluded the user interactions based on review information. **w/o_uu:** In this scenario, we excluded the user's trust network, meaning the model does not capture the knowledge-aware dependencies among users during training.

## 5.4 IMPACT OF THE HYPERPARAMETERS

The proposed method includes several adjustable parameters, such as the number of layers, final embedding dimensions, and learning rate, which must be optimized to achieve the best performance. To address the issue of over-smoothing common in GNNs, we experimented with different numbers of layers to evaluate their influence on the model's performance. To address Q3, we analyzed the experimental performance on the Epinions and Yelp datasets using different numbers of layers, as shown in Figure 3. The results indicate that the best performance for both datasets is achieved when the model uses two layers. The reason for this is that adding more layers leads to over-smoothing, a common issue in GNNs that becomes more pronounced as the number of layers increases. As the model performs repeated aggregation and transformation steps, the node representations tend to converge toward a uniform representation. This convergence reduces the model's ability to capture nuanced differences between nodes, ultimately hindering its performance.

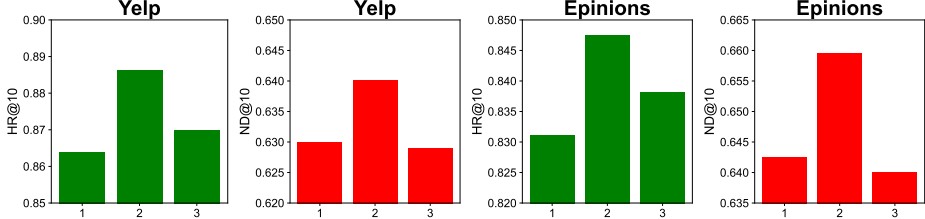

Figure 3: Evaluation of MvL-GNN over a different number of layers.

Figure 4 illustrates the performance of our method with respect to the final embedding dimension. The results show that performance begins to decline when the dimension exceeds 32. This decline can be attributed to the importance of selecting an appropriate dimension for effectively capturing semantic information. Increasing the dimension beyond a certain point may introduce redundant information, negatively impacting the model's generalization ability. Therefore, selecting the embedding dimension to achieve optimal performance is crucial.

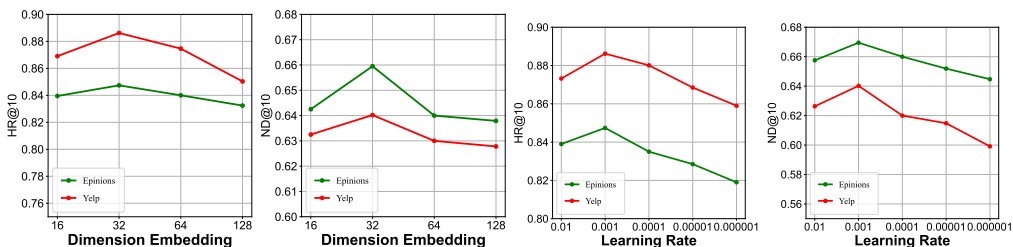

Figure 4: Evaluation of MvL-GNN over different dimension values and learning rates.

Figure 4 also illustrates the impact of different learning rates on the performance of MvL-GNN. The experiments reveal that increasing the learning rate initially leads to significant improvements in performance. However, once the learning rate exceeds 0.001, the model's performance on the Yelp and Epinions datasets declines. It is crucial to strike a balance, as setting the learning rate too high can cause unstable training and hinder convergence. At the same time, a learning rate that is too low may result in slow progress and the model getting stuck in suboptimal local minima.

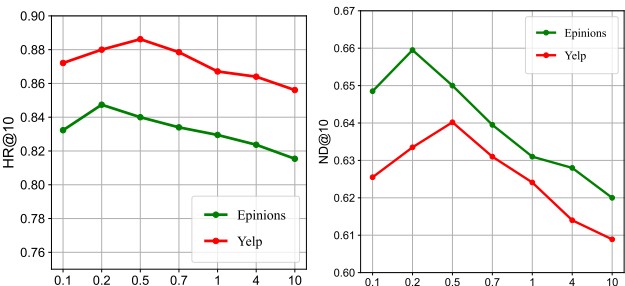

Figure 5: Impact of $\tau$ on model performance.

Figure 5, represents the impact of $\tau$ on the performance of the model. For the Epinions dataset, the optimal $\tau$ is between 0.1 and 0.5, while for Yelp, it is between 0.2 and 0.7. The performance on the Ciao dataset follows a similar trend to that of Epinions.

# 6 CONCLUSION

In this paper, we proposed MvL-GNN, a multi-view graph representation learning method specifically designed to address the limitations of traditional recommender systems. This method addresses the limitations of traditional approaches by incorporating multiple data sources, such as user trust relationships and user reviews, to enhance recommendation accuracy. The approach leverages user trust networks to capture implicit social influences. It integrates a pre-trained language model like BERT to transform user-generated reviews into meaningful embeddings, which helps capture nuanced user sentiments. Furthermore, the method employs contrastive learning to align and integrate diverse data representations, ensuring consistency and complementarity between the different sources of information. Through extensive experimentation on three real-world datasets (Yelp, Ciao, and Epinions), we demonstrated the effectiveness of MvL-GNN in outperforming state-of-the-art models across multiple metrics, including Hit Ratio and NDCG. Including multiple perspectives—user trust networks, item reviews, and meta-path-based item graphs—proved crucial in generating high-quality user and item embeddings, enhancing accuracy and robustness in recommendation tasks. Consequently, MvL-GNN has proven to be an effective solution for capturing the complex relationships between users and items in a recommendation setting, leveraging multiple sources of information to address common challenges such as data sparsity and cold start problems. Future work could explore further enhancements to MvL-GNN by incorporating additional data sources or experimenting with more advanced contrastive learning techniques.

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

# 7 APPENDIX

Table 3: Performance of NGCF and LightGCN methods on three datasets

| Datas | Metrics | NGCF | LightGCN | MvL-GNN | p-val |
|-------|---------|------|----------|---------|-------|
| Ciao | HR@10 | 0.6710 | 0.6801 | 0.7400 | 7e-9 |
| | ND@10 | 0.4612 | 0.4798 | 0.5301 | 8e-9 |
| Epinions | HR@10 | 0.7845 | 0.7985 | 0.8474 | 3e-6 |
| | ND@10 | 0.5845 | 0.5990 | 0.6595 | 3e-6 |
| Yelp | HR@10 | 0.7991 | 0.8079 | 0.8862 | 2e-5 |
| | ND@10 | 0.5700 | 0.5802 | 0.6402 | 1e-4 |

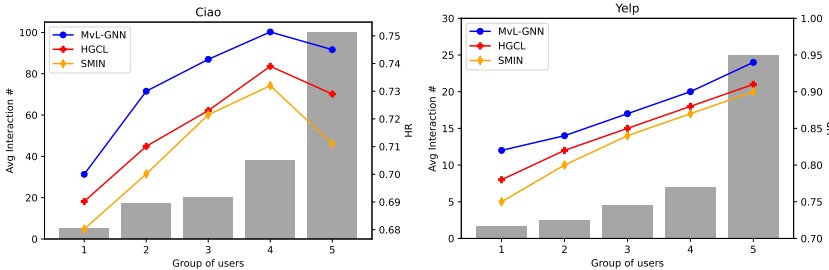

Figure 6: Performance comparison with respect to different data sparsity degrees on Yelp and Ciao datasets.

Figure 6, represents the effect of the MvL-GNN, HGCL and SMIN models in different sparsity. We divide the set of users into five groups to represent diverse user active degrees. The HR metric of each method is presented in the right side of y-axis. The left side y-axis represents the number of average number of interactions in each user group with bars. It is obvious that our proposed method out perform other two methods under different sparsity environments. The improvements of MvL-GNN comes from the integration of multiple sources of information, such as user trust relationships and user reviews and the use of a language model that allows the system to transform user-generated reviews into meaningful embeddings. Therefore, through the conducted experiments, MvL-GNN is able to maintain a decent performance even with sparse user-item interactions.

