# OpenReview forum: "Multi-View Graph Neural Networks with Language Models for Mutli-Source Recommender Systems"
_ICLR.cc/2025/Conference — Submitted to ICLR 2025_

### Official Review · Reviewer_ybKS · 2024-10-30

**Soundness:** 3
**Presentation:** 3
**Contribution:** 1
**Rating:** 3
**Confidence:** 5

**Summary:**

The paper introduces MvL-GNN, a multi-view graph neural network framework designed to enhance recommender systems by integrating diverse data sources, including user trust networks, item-wise meta-path based relationships, and user-generated textual reviews. Utilizing LightGCN for capturing user-item interactions, BERT for encoding textual reviews, and an attention mechanism to merge multiple views, MvL-GNN generates enriched embeddings for users and items. The framework employs contrastive learning to align and unify representations across different views, ensuring consistency and complementarity. Extensive experiments on real-world datasets such as Yelp, Ciao, and Epinions demonstrate that MvL-GNN significantly outperforms state-of-the-art methods in terms of Hit Ratio and NDCG, highlighting its effectiveness in addressing challenges like data sparsity and cold start problems in recommendation tasks.

**Strengths:**

1. **Comprehensive Integration of Multiple Data Sources:**
The paper presents a robust multi-view approach that incorporates user trust networks, meta-path based item interactions, and textual reviews into the recommendation framework. By leveraging these diverse data sources, MvL-GNN captures a more holistic view of user preferences and item characteristics, which potentially leads to higher recommendation accuracies compared to models that rely solely on user-item interactions.

2. **Effective Utilization of Language Models and Contrastive Learning:**
The integration of pre-trained language models such as BERT for encoding user and item reviews allows the model to extract rich semantic information from unstructured text data. Additionally, the use of contrastive learning mechanisms to align and unify representations across different views ensures that the embeddings are both consistent and complementary. The experimental results, showing significant performance improvements on multiple real-world datasets, underscore the effectiveness of incorporating textual features and advanced learning techniques to enhance recommendation accuracy.

**Weaknesses:**

1. **Limited Novelty:**
While the integration of multiple data sources is a notable aspect of the proposed method, the core components—namely, the use of language models for textual encoding and the application of contrastive learning techniques—are well-explored in recent literature (references [1-4]). The paper does not sufficiently highlight the fundamental differences, unique contributions, or novel innovations of MvL-GNN that set it apart from existing methodologies, thereby limiting the perceived novelty of the approach.

[1] Representation learning with large language models for recommendation
[2] Text is all you need: Learning language representations for sequential recommendation
[3] Heterogeneous graph contrastive learning for recommendation
[4] Self-supervised heterogeneous graph neural network with co-contrastive learning

2. **Insufficient Analysis of Performance Gains and Statistical Significance:**
Although the experiments demonstrate that MvL-GNN outperforms state-of-the-art methods across various datasets and metrics such as Hit Ratio and NDCG, the magnitude of these improvements may not be substantial enough to warrant the additional complexity introduced by the multi-view framework. Moreover, the paper lacks statistical significance tests to validate whether the observed performance gains are robust and not merely due to random variations, thereby weakening the strength of the experimental claims.

**Questions:**

1. **Scalability and Efficiency**: How is the model efficiency of the proposed methods, in comparison to the compared baselines?
2. **Concrete Examples for Using Textual Reviews to Enhance Recommendation**: Can the authors provide some examples on how textual review data benefits the recommendation accuracy?

---

### Official Review · Reviewer_JXBc · 2024-11-04

**Soundness:** 2
**Presentation:** 3
**Contribution:** 1
**Rating:** 3
**Confidence:** 4

**Summary:**

To ease the sparse user-item interaction problem in recommendation, the authors involve additional information sources (social trust and user reviews) for better modeling and propose a multi-view graph representation learning method. A pre-trained language models BERT is used to obtain review embeddings, and a GCN is used to encode additional knowledge from U-U/I-I networks that are built on the user’s trust network and item-item relationships. The experiments over three datasets show its effectiveness over the compared baselines. However, the novelty of the motivation of introducing addition information (such as user’s social network and reviews) and proposed model is weak. Besides, the experiments are also insufficient.

**Strengths:**

1)	The description of the paper is clear and easy to follow.
2)	The prediction accuracy over the compared baselines shows its effectiveness and the authors also perform ablation study to evaluate the effect of each component.

**Weaknesses:**

1)	The motivation of introducing addition information (such as user’s social network and reviews) for easing the user-item sparse interaction problem is well studied and lots of previous work in the cold-start and transfer learning domain are explored.
2)	The designed methods that deploys the BERT and GCN to encode textual review information and social network representation are widely-used in recommender system, also for contrastive learning. The novelty of the proposed model is weak.
3)	The experiments compare the proposed model with several baselines without additional information (social trust and user reviews), which is unfair. It is important to provide these information to the baselines as well.
4)	The improvement in Table 2 is too small given additional information used and no significant test were conducted. Besides, the comparison with backbone model lightGCN is missing.
5)	Insufficient analytical experiments and lack of experiments on the effectiveness of models under different sparsity.

**Questions:**

1)	What is the innovation of this paper and what is the difference from existing work?
2)	It is important to provide these information (social trust and user reviews) to the baselines and show their performance.
3)	What is the effect of the backbone model lightGCN?
4)	What is the effect of the model in different sparsity?

---

### Official Review · Reviewer_wTGU · 2024-11-04

**Soundness:** 2
**Presentation:** 3
**Contribution:** 2
**Rating:** 5
**Confidence:** 3

**Summary:**

This work proposes a multi-view GNN framework that integrates diverse information sources using contrastive learning and language models. The method employs a lightweight Graph Convolutional Network (LightGCN) on user-item interactions to generate initial user and item representations. An attention mechanism is utilized for the user view to incorporate social trust information with user-generated textual reviews, which are transformed into high-dimensional vectors using a pre-trained language model. Similarly, all reviews associated with each item are aggregated, and language models are used to generate item representations for the item view. An item graph is constructed by applying a meta-path to the user-item interactions. GCNs are applied to both the social trust network and the item graph, generating enriched embeddings for users and items. To align and unify these heterogeneous data sources, a contrastive learning mechanism is employed to ensure consistent and complementary representations across different views.

**Strengths:**

* The dataset and code have been provided, ensuring reproducibility.

* The article has a clear structure and is easy to follow.

**Weaknesses:**

* The innovation of this article is limited; techniques like LightGCN, attention mechanisms, item graphs based on meta-paths, and contrastive learning are several years old and should not be the main selling points of the current work.

* This article is too basic in terms of idea, method, and experiments, making it unsuitable for submission to ICLR.

* Such a basic work should have advantages in time complexity and runtime efficiency, but it appears to lack these, not aligning with the style of industry papers.

**Questions:**

* Do you have the latest baselines for heterogeneous relationships?

* What advantages does this work have compared to HGCL?

* Is the pre-trained language model specifically trained by you, or is it directly using the Transformers API with the original Hugging Face model?

---

### Official Review · Reviewer_dCsJ · 2024-11-05

**Soundness:** 2
**Presentation:** 3
**Contribution:** 2
**Rating:** 5
**Confidence:** 3

**Summary:**

This paper introduces MvL-GNN, a model that leverages user-user and item-item graphs to generate two additional contrastive views for contrastive learning, thereby improving model performance.

**Strengths:**

1. This paper is easy to follow, and the proposed method is clearly presented.
2. The experiments conducted appear to demonstrate the effectiveness of the proposed method.

**Weaknesses:**

1. There is a lack of performance comparison with methods that only consider the user-item graph, such as LightGCN and NGCF.
2. There is a lack of significance analysis (e.g., p-value) regarding the performance improvement of the proposed method compared to the baselines.

**Questions:**

See Weaknesses.

---

### Official Review · Reviewer_v4GY · 2024-11-05

**Soundness:** 2
**Presentation:** 3
**Contribution:** 2
**Rating:** 3
**Confidence:** 5

**Summary:**

This paper studies the problem of enhancing recommender systems by incorporating multi-source data to improve recommendation accuracy and robustness. The authors propose a multi-view graph neural network framework that leverages contrastive learning and pre-trained language models to integrate user-item interactions, social trust networks, and user reviews.

**Strengths:**

S1. The majority of the paper is easy to follow.

S2. The proposed multi-view framework effectively integrates diverse information sources, including user-item interactions, trust networks, and review data.

**Weaknesses:**

W1. I have some concerns regarding the motivation. The inclusion and processing of raw textual data (such as user reviews) inherently add complexity and computational cost to the recommendation system, particularly in large-scale datasets. It would benefit the paper to include a detailed cost-benefit analysis to justify the addition of textual information.

W2. The baseline models do not incorporate review data, which raises questions about the fairness of the comparisons. It is difficult to discern whether the observed performance improvements caused by the added information or the multi-view design itself. Additionally, the performance gains achieved seem marginal.

W3. In Table 1, the number of nodes and edges in the Epinions dataset is notably lower than in prior studies. The authors should detail the data processing approach for Epinions to clarify the reasons for this discrepancy.

W4. The presentation of the paper needs some improvements. For example, there are several typos in the conclusion.

**Questions:**

See W1-W4.

---

### Meta-Review · Area_Chair_nKfh · 2024-12-20

**Metareview:**

This paper addresses the challenge of improving recommender systems by integrating data from multiple sources to enhance both recommendation accuracy and robustness. The authors introduce a multi-view graph neural network framework that combines contrastive learning with pre-trained language models to effectively incorporate user-item interactions, social trust networks, and user reviews.

There are some strengths of the paper:
(1) The paper is well written
(2) The conducted experiments can well demonstrate the effectiveness of the proposed method.

However, the reviewers also pose many significant weaknesses of the paper:

(1) The motivation for incorporating review data.
(2) The lack of performance comparison with many Sota methods.
(3) The novelty of the model is not well justified.
(4) The scalability of the proposed method is not well discussed.

In summary, I believe the paper still needs further revisions before it can be published.

**Additional Comments On Reviewer Discussion:**

In the rebuttal period, the reviewers mainly raise concerns about the motivation, experiments,  and scalability of this paper. The reviewers can solve parts of the reviewers' concerns. However, many details are missing. For example, how the datasets are processed, such that the data statistics are different from the original one.

---

### Decision · Program_Chairs · 2025-01-22

Reject